# Genetic and Transcriptome Analysis of Leaf Trichome Development in Chinese Cabbage (*Brassica rapa* L. subsp. *pekinensis*) and Molecular Marker Development

**DOI:** 10.3390/ijms232112721

**Published:** 2022-10-22

**Authors:** Jingjuan Li, Hongxia Wang, Dandan Zhou, Cheng Li, Qian Ding, Xiaogang Yang, Fengde Wang, Han Zheng, Jianwei Gao

**Affiliations:** 1Shandong Branch of National Vegetable Improvement Center, Institute of Vegetables, Shandong Academy of Agricultural Sciences, Jinan 250100, China; 2College of Life Science, Shandong Normal University, Jinan 250100, China

**Keywords:** Chinese cabbage, genetic analysis, molecular marker, transcriptome analysis, trichome development

## Abstract

Chinese cabbage (*Brassica rapa* L. subsp. *pekinensis*) is one of the vegetables with the largest cultivated area in China and has been a great addition to the daily diet of Chinese people. A genetic map has been constructed in our previous study using the F_2_ population of two inbred lines of Chinese cabbage, namely “G291” (a hairy line) and “ZHB” (a hairless line), based on which a candidate gene related to trichome traits was identified on chromosome A06 with a phenotypic variance of 47%. A molecular marker was found to co-segregate with the trichome traits of the F_2_ population, which is in the 5′-flanking region of *BrGL1*, and a corresponding patent has been granted (NO. CN 108545775 B). Transcriptome analysis was carried out on the cotyledon, the first true leaf and the leaf closest to each inflorescence of F_2_ individuals of “G291 × ZHB” with or without trichomes, respectively. Ten pathways, including 189 DEGs, were identified to be involved in the development of trichomes in Chinese cabbage, which may be specifically related to the development of leaf trichomes. Most of the pathways were related to the biosynthesis of the secondary metabolites, which may help plants to adapt to the ever-changing external environment. DEGs also enriched the “plant-pathogen interaction” pathway, which is consistent with the conclusion that trichomes are related to the disease resistance of plants. Our study provides a basis for future research on the occurrence and development of trichomes in Chinese cabbage.

## 1. Introduction

Trichomes are hair-like structures extending from epidermis cells on the surfaces of different organs of plants, including leaves, stems and floral organs. As a natural physical barrier between the plant and its external environment, trichomes play a crucial role in protection from biotic and abiotic stresses, such as insect herbivores, pathogenic microorganisms, temperature, water, ultraviolet (UV) radiation, and heavy metals [1,2,3]. They are also related to the characteristic textures of different vegetables [4]. Revealing trichome development mechanisms in non-model species is of great significance to morphogenesis and efficient breeding strategy.

Because the trichomes in *Arabidopsis thaliana* are typically single-celled and non-glandular, the molecular mechanisms of trichome initiation and development have been well studied, and a number of genes have been characterized in this model plant. A lateral inhibition mechanism is employed in the formation of trichomes in *Arabidopsis* [5] through the interaction of a series of transcription factors, which have been shown to play positive or negative roles in plant growth and development. Typical positive regulators include *GLABRA1* (*GL1*), *GL2*, *GL3* [6,7,8], *ENHANCER of GLABRA3* (*EGL3*) [9], *Transparent Testa*
*Glabra 1* (*TTG1*) [10], *GLABROUS*
*INFLORESCENCE STEMS* (*GIS*), *GIS2*, *GIS3* [11,12,13], and Zinc Finger Protein 5, 6, and 8 [12,14,15], and typical negative regulators include *Triptychon* (*TRY*) [16], *CAPRICE* (*CPC*) [17], *ENHANCER of TRY and CPC1* (*ETC1*), *ETC2* [18,19], *TRICHOMELESS1* (*TCL1*) [20], and *SQUAMOSA PROMOTER**-BINDING*
*PROTEIN-LIKE* (*SPL*) [21]. *GL1*, a member of the R2R3-MYB family, is the first cloned gene related to the development of trichomes, which acts specifically in trichome patterning [22]. GL1, TTG1 and GL3 and EGL3 form a trimeric complex, namely MBW (R2R3MYB, basic helix-loop-helix, and WD40). MBW acts as a transcriptional activation complex to trigger the expression of *GL2*, which is believed to be the activator of trichome-specific differentiation genes, leading to the promotion of trichome formation [23]. Simultaneously, the MBW protein complex can repress the expression of some R3 MYB genes. Proteins encoded by these *R3 MYB* genes are transferred from hair precursor cells to adjacent cells, which block the formation of the MBW complex by competitive binding with GL3 or EGL3, and, thus, finally inhibit the formation of trichome [22,24,25]. The negative regulators, such as TRY, CPC, and ETCs, inhibit the initiation of trichomes by competing with GL1 in binding with GL3 [17,18,24,25].

Both similarities and differences have been observed in the occurrence and regulation mechanisms of trichomes in different non-model plants. *Gossypium hirsutum MYB2* (*GhMYB2*), a putative *GL1* homolog, and one of its downstream genes, *GhRDL1*, promote hair formation of seeds when co-overexpressed in *A. thaliana* Columbia-0 (Col-0) wild type plants, while co-overexpression of these two genes in *A. thaliana* Col-0 *try* mutant plants produce both silique trichomes and seed hairs [26]. However, overexpression of the *Arabidopsis GL1* gene in tobacco has no effect on the trichome phenotype of the tobacco plants. When *CotMYBA*, an *MYB* gene family member in *Gossypium hirsutum* ovules, is overexpressed in Arabidopsis and tobacco, cotyledonary trichomes are observed in tobacco transformants, while the production of trichomes shows no significant change in *Arabidopsis* [27]. These results implicated that trichome formation and regulation mechanisms are specific and unique in different species.

Chinese cabbage is one of the vegetables with the largest cultivated area in China. Trichomes cover nearly the whole body of Chinese cabbage except cotyledons and epicotyls. Since humans prefer leafy vegetables with few or no trichomes, it is important to understand the molecular genetic mechanisms of trichome development.

Quantitative trait loci (QTL) analysis is an effective tool to identify candidate genes responsible for important agronomic traits related to *B. rapa* yield. Most of these traits are quantitative traits that are regulated by one or more major genes under different internal or/and external environments [28]. Using molecular markers, such as amplified fragment length polymorphisms (AFLP), restriction fragment length polymorphisms (RFLP), simple sequence repeat (SSR) markers, and single nucleotide polymorphisms (SNP), a number of genetic linkage maps have been constructed in different genetic populations of Chinese cabbage [29,30,31,32]. These studies have identified many important QTLs related to seed coat color, reproductive fitness traits, orange inner leaves and flowering time. Many studies have also been carried out on the genetic features of the trichomes in Chinese cabbage, which show that trichomes are a quantitative phenotype collectively controlled by a number of genes. *Bra009770*, located on chromosome A06, is highly homologous to TTG1 in *Arabidopsis* and closely associated with the development of leaf hairs and seed coat color [33]. Li et al. (2011) [34] found that a 5-bp deletion on exon 3 of *BrGL1*, located on chromosome A06, is responsible for the hairless trait of Chinese cabbage. In another report by the same authors in 2013, they discovered that the large variation of the 3′-terminal of *GL1a* may be critical for leaf hair traits of the hairless and hairy lines of *Raphanus sativus* [5]. Simple sequence repeat (SSR) markers were analyzed using a large-scale F_2_ population, based on which *Bra025311*, a close homolog to *GL1* of *Arabidopsis thaliana*, was identified as a candidate gene controlling the trichome trait in Chinese cabbage [35]. QTL analysis was performed on a hairy line, Mibuna, and a hairless line, Mizuna, of Chinese cabbage. The sequences of the 3′-terminal of *BraA09g003340*, also homologous to *GL1*, are considerably different between these two lines, which may contribute to the variation of the trichome phenotypes [4].

Here, QTL and co-segregation analysis were carried out to identify candidate genes involved in the development of trichomes based on SSR markers and a large-scale F_2_ population of a hairy line “G291” and a hairless line “ZHB” of Chinese cabbage. RNA-Seq was used to reveal the related pathways of leaf trichome formation and development. Understanding the mechanisms of trichome development and the functions of candidate genes will be helpful in cultivating new varieties of hairless Chinese cabbage.

## 2. Results

### 2.1. Genetics of Leaf Trichome

Two inbred lines of Chinese cabbage, “G291” (a hairy line) and “ZHB” (a hairless line), were used in this research. “G291” has a large number of trichomes, whereas “ZHB” has few trichomes on the surface of its leaves (Figure 1).

An F_2_ segregating population was constructed using “G291” and “ZHB” as parents. A total of 1456 healthy plants, including 1075 hairy plants and 381 hairless plants, were obtained from 1500 seeds of the F_2_ generation (Table 1). The χ^2^ test showed that the segregation ratio was in accordance with the expected Mendelian inheritance ratio, indicating that the trichome trait may be a quality characteristic that is controlled by a dominant gene.

### 2.2. Mapping of the Candidate Gene and the Development of Molecular Markers

In our previous study [36], a genetic map was constructed based on the two inbred lines of Chinese cabbage, “ZHB” and “G291”, and their F_2_ population. Based on this genetic map, genes related to trichomes were located on chromosome A06 with a phenotypic variance (PVE) of 47% (Table 2). Here, the candidate gene was designated as *Bra06H*.

Using the phenotypic data and the genetic linkage map, *Bra06H* was primarily located on chromosome 6 between the SSR markers A6S38 and A6S39. The physical length is 1.28 Mb (between 20,377,076 and 21,661,391 bp of chromosome 6), which contains 360 genes.

Based on the sequence between A6S38 and A6S39 from the BRAD database, 50 more pairs of SSR primers were designed for fine mapping. Electrophoresis results showed that five of these primer pairs produced polymorphic sequences, including A6S75, A6S76, A6S95, A6S103, and A6S82. Thus, *Bra06H* was localized between marker A6S95 and A6S103 in a region that was 15 Kb long (Figure 2a).

Among this region, a molecular marker (MK-F: TTCCATACCTCCCTCACTGGTA, MK-R: TTCAACTGTCCACAACCCTTTC) linked and co-segregated with the trichome traits of these two parent lines. We verified this molecular marker in the F_2_ population (Figure 2b) and commercial varieties of Chinese cabbage (Figure 2c). These results showed that MK was completely linked to the leaf hair phenotype and could be applied to the breeding of new varieties of hairless Chinese cabbage.

Electrophoresis results showed that the product in the hairless F_2_ individual was 144 bp long, which is the same as that of the hairless parent ZHB. Additionally, the product from G291 was 156 bp long. The length of PCR products between the two groups could be distinguished by denaturing RAGE. All the hairless individuals were homozygous. However, in the hairy group, there were two genotypes: one was the homozygous type, similar to the hairy parent G291, while the other was the heterozygous type containing two parental bands. The molecular marker has been patented (published number: CN 108545775 B).

We cloned the sequences from G291 and ZHB using the MK primers, where a 12-bp indel site was identified (Figure 2d). It was in the 5′-flaking region of *BrGL1*, encoding an R2R3MYB transcription factor, which plays an important role in regulating the occurrence and development of leaf trichomes in *Arabidopsis* and *Brassica*. It also indicated that *BrGL1* might be the dominant gene controlling trichome traits in this population.

### 2.3. Sequence Analysis of the BrGL1gene

We cloned the whole-length DNA sequences of BrGL1 from G291 and ZHB with the primers BrGL1-Seq (F: ATGAGAACGAGGAGAAGAACAGAGG; R: CTAGAGGCAGTAGCCAGTATCACCGGAC). Sequence alignment analysis (Figure 3) showed that the BrGL1 DNA sequences from G291 and ZHB contained 3 exons and 2 introns. Compared with the sequence of BrGL1 published in the BRAD database (chiffu), 11 SNPs and 1 indel site were found in the exons of BrGL1 from G291 and ZHB. The sequences of the second intron of BrGL1 from G291 and ZHB were significantly different from that of the predicted sequence. At the DNA level, there was an SNP and an indel site between G291 and ZHB, but the only frame-shift mutation was a 5-bp deletion in the third exon of ZHB, which may be responsible for the absence of leaf trichomes in ZHB.

### 2.4. Expression Patterns of BrGL1 in Various Tissues of the Two Parent Breeds

qRT-PCR was used to explore the differential expression level of *BrGL1* in diverse tissues of “ZHB” and “G291” (Figure 4). These results showed that the expression levels of *BrGL1* in most tissues of G291 were higher than those of ZHB. In G291, the highest expression level of *BrGL1* was in the root, followed by the stem, and the lowest was in the leaf. In the flower bud, root and silique, the expression levels of *BrGL1* in G291 were almost 2, 4, and 27 times higher than those in ZHB, respectively. These two lines of Chinese cabbage had a similar expression level of *BrGL1* in the stem.

### 2.5. Transcriptome Analysis

The hairy and hairless genotypes of the F_2_ population constructed from a cross between “G291” and “ZHB” were sampled to establish a mixed pool with trichomes (T) and a mixed pool without trichomes (NT), respectively. Cotyledons (C), the first true leaves (E), and stem leaves (S) of 20 plants from each pool were collected for RNA extraction, respectively. Equal quantities of the total RNA were pooled for cDNA library construction. Each sample had three biological replicates, and eighteen libraries were generated.

A total of 18 samples were sequenced using the BGISEQ-500 platform, with an average of 23.92 M reads per sample. Low-quality, adaptor-polluted and high content of unknown base (N) reads were removed before downstream analyses. Then, clean reads were mapped to the reference genome and the reference transcripts using HISAT and Bowtie 2, respectively. The average mapping ratio with the reference genome was 91.10%, and the average mapping ratio with the reference gene was 77.41%, as shown in Table 3. A total of 35,305 genes were detected. The expression levels of the genes were quantified by RSEM. We calculated the number of genes under three different FPKM ranges (FPKM ≤ 1, FPKM 1~10 and FPKM ≥ 10), as shown in Appendix A.

Pearson correlation coefficients for all gene expression levels between each sample pair were calculated to reflect the gene expression correlation between samples and are shown in the form of heat maps (Appendix A). The correlation coefficients of the three biological replicates of each sample were higher than 0.9, indicating that the samples had great repeatability.

### 2.6. Screening of DEGs and Clustering Analysis

The DEGSeq method was used to find differentially expressed genes between groups. The DEGs were selected according to the following default criteria: log_2_ (fold change) > 1 or <−1 and Q-value < 0.001. A total of 17,677 DEGs were screened out from all 35,305 genes, and 804 DEGs were found in all four paired comparisons (Figure 5a).

In a comparison of the gene expression level in the hairless groups (NTC vs. NTE, NTS vs. NTE), there were 3095 and 1719 DEGs expressed higher in NTC and NTS than in NTE, while 1407 and 5648 DEGs were expressed lower in NTC and NTS than in NTE, respectively (Figure 5b). In the hairy groups, the expression levels of 2781 and 6729 DEGs were up-regulated, while 3896 and 3448 DEGs were down-regulated in TC and TS compared with those in TE, respectively. Compared with cotyledons of hairless materials (NTC), 2752 genes were up-regulated, and 244 genes were down-regulated in the cotyledons of hairy materials (TC). Comparing the true leaf of hairless materials (NTS) with that of hairy materials (TS), 3037 genes were up-regulated, and 1287 genes were down-regulated in the true leaf (TE) of the hairy materials. Between the stem leaves of hairless and hairy materials (NTS vs. TS), 891 genes were up-regulated, and 3587 genes were up-regulated in TS (Figure 5b).

The expression trends of the 804 DEGs shared by the four pairs in the Venn diagram were analyzed by the OmicShare tools, a free online platform for data analysis (http://www.omicshare.com/tools (accessed on 21 October 2021)). Among 20 sub-clusters, there were four distinct sub-clusters with an extremely low *p*-value, containing 201, 93, 137 and 229 DEGs, respectively (Figure 5c). The numbers of genes within each cluster were marked on each profile (Figure 5d). These gene clusters with a similar expression trend may play an important role in the development of leaf trichomes in Chinese cabbage, especially sub-clusters 3 and 16, in which DEGs had opposite expression trends in the hairy and hairless groups.

These results show that different DEGs were expressed at different stages of plant growth and development, which might be involved in the leaf trichome formation of Chinese cabbage.

### 2.7. GO and KEGG Pathway Analysis

Annotation analysis of GO was performed to understand the distribution of gene functions of Chinese cabbage. According to the numbers of DEGs enriched to the same GO term, the results of GO functional enrichment among the 9 comparison groups were similar. In terms of molecular function, the top three GO terms were cellular processes, metabolic processes and single organization processes. In terms of cell components, the terms were mainly concentrated in the cell, cell part and organelle. In terms of biological processes, the terms were mainly involved in binding and catalytic activity. The GO terms with a corrected *p*-value < 0.01 were analyzed, and the results showed that there were 277 GO terms, including 50 terms belonging to cellular components, 144 terms to biological processes and 83 terms to molecular functions (Appendix A). In the hairless group, the enrichment of DEGs of NTC vs. NTE and NTC vs. NTS was relatively similar to GO terms, including 10 in the cellular components, 27 in biological processes and 8 in molecular function. However, the DEGs of NTE vs. NTS were enriched to 12 terms in the cellular component, which were significantly different from any other GO terms in the cellular component enriched by DEGs of NTC vs. NTE and NTC vs. NTS. DEGs of NTE vs. NTS were specifically enriched to 14 terms in biological processes and 16 terms in molecular function. For the hairy groups, DEGs of TC vs. TE, TC vs. TS and TS vs. TE were enriched to 4, 13, and 13 terms in cellular components, respectively, and enriched to many terms in biological process and molecular function. These results may be related to the different functions of leaves in different developmental stages of plants but may not be related to the development of leaf trichomes. By comparing the hairless and hairy groups during the same period, it was found that the DEGs of NTC vs. TC were enriched to 4 terms in cellular components, 27 in molecular function and 24 in biological processes, while DEGs of NTE vs. TE were enriched to 10 in cellular components, 15 in molecular function and 36 in biological processes. DEGs of NTS vs. TS were enriched to 8 terms in cellular components, 5 in molecular function and 20 in biological processes. Considering that only the sample TE in the hairy group has trichomes, we focused on the GO terms enriched by DEGs of NTE vs. TE. There were 5 terms in molecular function, including “oxygen binding”, “transcription regulatory region DNA binding”, “regulatory region DNA binding”, “regulatory region nucleic acid binding” and “sequence-specific DNA binding”, and 8 terms in biological process, including “response to organic substance”, “response to hormone”, “response to osmotic stress”, “response to salt stress”, “response to lipid”, “response to abscisic acid”, “response to alcohol” and “response to jasmonic acid”. However, these GO terms contained a considerable number of DEGs, and GO analyses did not ideally narrow down the range of the candidate genes.

Then, we analyzed the enrichment pathways with a level of Q value ≤ 0.01 (Table 4).

We found that there were 3 common pathways enriched by DEGs of NTC vs. NTE and TC vs. TE, including “Biosynthesis of secondary metabolites”, “Pentose and glucuronate interconversions,” and “Plant hormone signal transduction”.

Between pathways enriched by DEGs of NTE vs. NTS and TE vs. TS, two common pathways, “Photosynthesis—antenna proteins” and “Starch and sucrose metabolism”, were identified. Pathways enriched by DEGs from NTC vs. NTS and TC vs. TE included “Biosynthesis of secondary metabolites”, “Photosynthesis-antenna proteins”, “Carbon metabolism”, “alpha-Linolenic acid metabolism,” and “Photosynthesis”. These pathways may be related to the development of plant leaves, or these pathways generally exist in the process of leaf development but may not be specifically related to the development of leaf trichomes. Besides these common pathways, the DEGs of NTE vs. TE were also enriched in 10 other pathways, including “Metabolic pathways”, “Benzoxazinoid biosynthesis”, “Limonene and pinene degradation”, “Stilbenoid, diarylheptanoid and gingerol biosynthesis”, “Sesquiterpenoid and triterpenoid biosynthesis”, “MAPK signaling pathway-plant”, “Indole alkaloid biosynthesis”, “Circadian rhythm-plant”, “Plant-pathogen interaction” and “Nitrogen metabolism”.

According to the transcriptome analyses, a hypothesis model of *BrGL1* regulating leaf trichomes development in Chinese cabbage is shown in Figure 6.

### 2.8. Validation of the RNA-Seq Results by qRT-PCR

A total of 189 DEGs were identified in these 10 pathways, among which 102 DEGs were up-regulated with a log_2_Ratio (TE/NTE) ≥ 2 and 87 were down-regulated with a log_2_Ratio (TE/NTE) ≤ −2 (Appendix A). Six DEGs from the up- and down-regulated groups were selected for the validation of the RNA-seq data using qRT-PCR. The relative expression measurements from qRT-PCR are shown in Figure 7, and the qRT-PCR primers are listed in Appendix A. The results showed that the DEGs had similar expression patterns, indicating that the RNA-seq data could be used for the gene expression profiling of trichome traits of Chinese cabbage.

## 3. Discussion

Many important yield-related agronomic traits vary greatly among different Chinese cabbage varieties after a long period of evolution and breeding. QTL analysis, as a powerful tool, has been performed to identify candidate genes responsible for the morphological diversity of Chinese cabbage [5,37,38,39]. Different molecular markers, such as AFLP, RFLP, STS, SSR, InDel, and SNP, were used to construct genetic maps. Intragenic SSRs, especially the expressed sequence tags SSRs (EST-SSRs) in the transcriptional sequences, have been widely used in genetic linkage map construction in plants because these SSRs are more conserved and transferable than extragenic SSRs [40,41,42,43]. In our previous study [36], a genetic map was constructed using 105 co-dominant polymorphic SSRs in 240 individuals of the F_2_ segregating population crossed between two Chinese cabbage inbred lines, ZHB and G291, from which a candidate gene, *BrGL1*, was located on chromosome A06.

*BrGL1* (*Bra025311*), encoding an R2R3-type MYB transcription factor, was localized to a 0.55-Mb long region using fine mapping. A 5-bp deletion in the third exon of this gene in the hairless line ZHB resulted in a frame-shift mutation, which was consistent with previous studies [34,35]. However, the expression pattern of *BrGL1* in different tissues of the two Chinese cabbage lines was significantly different from previous studies. Ye et al. [35] found that the highest expression level of *Bra025311* is in the leaf, and it is not expressed in the leaves of the hairless line “FT,” which is consistent with the trichome location. In this study, *Bra025311* was expressed the highest in the root of the two lines of Chinese cabbage, and it was expressed at an extremely low level in the leaf, especially the leaves of the hairy lines G291. In 2011, Li et al. [34] reported that in mature leaves, where trichome development is completed, no transcript of *BrGL1* was observed by RT-PCR, which is in accordance with that in *A. thaliana* [44]. These results indicate that the network and mechanism of *BrGL1*, by which the leaf trichome traits of Chinese cabbage are regulated and complex, and there may be temporal and spatial differences between its expression and trichome development. Considering the high expression of *BrGL1* in the roots of the two varieties of Chinese cabbage, we speculate that *BrGL1* may be related to the development of root hairs, which needs to be further investigated.

Interestingly, the 5′-flanking region (upstream) of *BrGL1* has a 12 bp-deletion in the hairless line ZHB compared with that in the hairy line G291 (Figure 2d). A co-dominant indel marker based on the sequence differences of the 5′-flanking region of *BrGL1* between varieties with and without trichomes was developed and verified by a number of commercial varieties of Chinese cabbage with or without trichomes (Figure 2c). It is reported [4] that a *GL1* ortholog on A09 controls the trichome number in two Japanese leafy vegetables belonging to *Brassica rapa*, and the length of the 3′-flanking region (downstream) is the difference between these two germplasms. In *Arabidopsis*, the 3′-flanking region of *GL1* is important for the expression of *GL1* [44]. The sequence differences of this region may lead to different expressions of *BrGL1* on A09.

Considering that metabolites play important roles in plant growth and development, and the KEGG pathways are often used to describe the metabolic pathways in cells [45], we focus more on the analysis of the KEGG pathways. Besides the pathways generally in the process of leaf development, 10 pathways (Table 4) containing 189 DEGs (Appendix A) were identified, which may be specifically related to the development of leaf trichomes. DEGs were enriched to the “Plant-pathogen interaction” pathway, which is consistent with the conclusion that trichomes are related to the disease resistance of plants. We found that 6 out of the 10 pathways were related to the biosynthesis of secondary metabolites, which can help plants to adapt to the ever-changing external environment [46]. Transcriptome profiling of pepper was carried out to identify genes responsible for the formation of trichomes [47]. Among the top 20 KEGG pathways, 4 pathways are also identified in our study, including “Stilbenoid, diarylheptanoid and gingerol biosynthesis”, “MAPK signaling pathway-plant”, “Plant-pathogen interaction” and “Sesquiterpenoid and triterpenoid biosynthesis”. In a transcriptome profile of the petal trichome trait in *Lily* [48], only one common pathway, “Sesquiterpenoid and triterpenoid biosynthesis”, was found to be related to trichome traits. These results also indicate that the development of trichomes is regulated by complex molecular mechanisms. There may be a common mechanism for the development of trichomes in plants, while specific regulatory mechanisms may also exist between species.

## 4. Materials and Methods

### 4.1. Plant Materials

Two inbred lines of Chinese cabbage, “G291” (a hairy line) and “ZHB” (a hairless line), were used to construct the mapping population. These two lines are significantly different in both size and shape.

### 4.2. Development of Mapping Population and Phenotypic Statistics

QTL mapping of candidate loci for controlling trichome development in Chinese cabbage was performed using the genetic map constructed in our previous study [36].

An F_2_ population constructed from a cross between “G291” and “ZHB” was used for mapping. A total of 240 F_2_ individuals were planted in Changqing, Jinan, China, with a row placing of 50 cm and a plant placing of 50 cm. The detailed description of the mapping population and methods were according to Gao et al. (2022) [36].

In order to identify the segregation ratio of trichome traits of the F_2_ population, 1500 seeds were planted and grown in a greenhouse at 20 ± 2 °C with a 16 h light/8 h dark photoperiod. The trichome phenotypes were determined by examining the leaf hairs at the two-leaf stage.

### 4.3. DNA Extraction, PCR Condition and 8% Denaturing PAGE Electrophoresis

The total DNA of the parental lines and F_2_ individuals was extracted from young leaves using the sodium dodecyl sulfate (SDS) method [49] and quantified by calculating the absorbance ratio at 260 nm and 280 nm using a NanoDrop ND-1000 spectrophotometer (Nano-Drop, Wilmington, DE, USA). The DNA was diluted to 50 ng/μL and amplified by polymerase chain reaction (PCR). The PCR conditions were: 94 °C for 5 min; 36 cycles of 94 °C for 30 s, 55 °C for 30 s, 72 °C for 30 s, followed by 72 °C for 10 min. PCR products were electrophoresed on an 8% denaturing PAGE gel and visualized with silver staining [50]. SSR primers are listed in Appendix A.

### 4.4. Amplification and Sequence Analysis of BrGL1 in G291 and ZHB

The primers were designed using Primer 6.0. CDS and promotor sequences of *BraGL1* in “G291” and “ZHB” were amplified using KOD-Plus (TOYOBO) following the manufacturer’s instructions. The PCR conditions were: 95 °C for 3 min, followed by 32 cycles of 95 °C for 30 s, Tm (according to primers) for 30 s and 68 °C for 30 s, followed by 68 °C for 10 min.

Sequences were aligned using the DNAman (version 6) software.

### 4.5. Total RNA Isolation

A total of 300 seeds of the F_2_ population were germinated in a glass petri dish at 20 ± 2 °C for 24 h. Then, the petri dish was placed in a 4 °C fridge for 15 days for vernalization. The seedlings were transferred into pots and grown in the greenhouse at 20 ± 2 °C with a 16-h-light/8-h-dark photoperiod. At the two-leaf stage, the cotyledon and the first true leaf of the hairless group (20 seedlings in each biological replicate) were collected and named NTC and NTE, respectively. Similarly, the cotyledon and the first true leaf of the hairy group (20 seedlings in each biological replicate) were collected and named TC and TE, respectively. During the flowering stage, the leaf closest to each inflorescence was selected and named NTS and TS for the hairless group and hairy group, respectively. Three biological replicates were used for each sample. The total RNA was extracted from each sample using the Trizol reagent (Invitrogen, Carlsbad, CA, USA) and treated with RNase-free DNase I (Takara, Dalian, China). Equal quantities of the total RNA from three replicates were mixed for the cDNA library construction.

### 4.6. cDNA Library Construction, Sequencing and Data Processing

cDNA libraries were constructed at the Beijing Genomics Institute (BGI, Shenzhen, China) according to the manufacturer’s protocol (Illumina, Inc., San Diego, CA, USA). An Agilent 2100 Bioanalyzer (Agilent Technologies, Inc., Santa Clara, CA, USA) and an ABI Step OnePlus Real-Time PCR System (Applied Biosystems, Inc., Foster City, CA, USA) were used to qualify and quantify the sample library. The library products were sequenced via the BGISEQ-500 platform. Raw reads after quality control (QC) were filtered into clean reads and then aligned to the reference sequences. The alignment data were used to calculate the distribution of reads on the reference genes and the mapping ratio. Bowtie2 [51] was used to map clean reads to the reference gene, and HISAT [52] was used to map to the reference genome.

### 4.7. Gene Quantification, Screening DEGs, GO and KEGG Analysis

RSEM [53] was used to calculate the expression level of genes, by which the influence of different gene lengths and sequencing discrepancy on gene expression can be eliminated. DEGSeq [54] was used to identify differentially expressed genes (DEGs) according to the following default criteria: fold change ≥ 2 or ≤−2 and Q-value ≤ 0.001. All DEGs were mapped to the Gene Ontology (GO) terms in the Gene GO knowledgebase at http://www.geneontology.org (accessed on 25 January 2018). The GO terms with a *p*-value ≤ 0.01 are defined as the significantly enriched GO terms in DEGs. Pathway enrichment analysis of DEGs was performed using the Kyoto Encyclopedia of Genes and Genomes (KEGG) (http://www.genome.jp/kegg/ (accessed on 25 January 2018)).

### 4.8. qRT-PCR Verification

First-strand cDNA was synthesized using the PrimeScript 1st Strand cDNA Synthesis Kit (Takara, Shiga, Japan). Quantitative real-time PCR (qRT-PCR) was carried out using an SYBR Green Master mix (Takara) on an IQ5 Real-Time PCR Detection System (Bio-Rad, Hercules, CA, USA). The qRT-PCR primers designed for the candidate DEGs and *BrACT1* [55] are listed in Appendix A. The PCR cycling conditions comprised an initial polymerase activation step at 95 °C for 1 min, followed by 40 cycles of 95 °C for 10 s and 60 °C for 30 s. After each PCR run, a dissociation curve was designed to confirm the specificity of the product and to avoid the production of primer dimers. Three replicates of each sample were conducted to calculate the average Ct. The relative expression level was calculated by the comparative 2^−ΔΔCt^ method [56].

## 5. Conclusions

Our study shows that the 5 bp-deletion in the coding region and the 12 bp-deletion in the 5′-flanking region of *BrGL1* in the hairless line might cause the difference in trichome traits between the two Chinese cabbage varieties ZHB and G291. A co-dominant indel marker was developed (patent number: CN 108545775 B), which may be used for the rapid breeding of hairless Chinese cabbage varieties. A number of DEGs were identified by RNA-Seq as candidate genes regulating trichome development in Chinese cabbage. Further investigations are required to identify other factors related to trichome development.

## Figures and Tables

**Figure 1 ijms-23-12721-f001:**
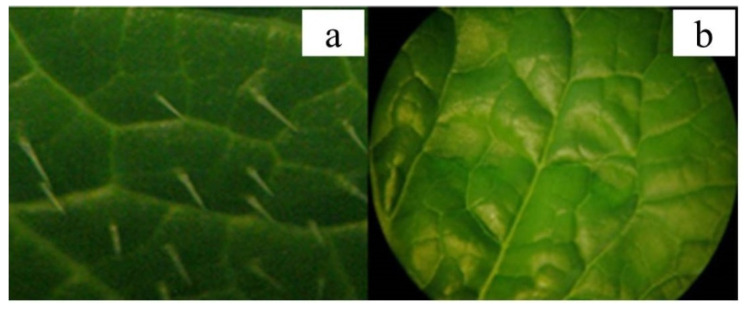
Morphological characteristics of Chinese cabbage leaves. (**a**) G291 (hairy line) and (**b**) ZHB (hairless line).

**Figure 2 ijms-23-12721-f002:**
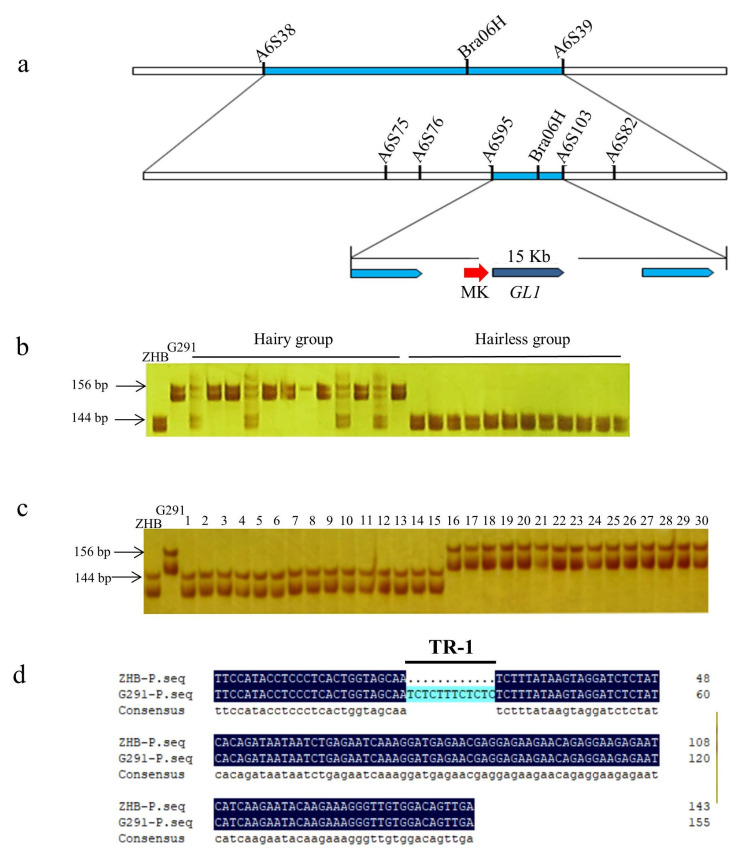
Genetic mapping of trichome candidate gene and development of molecular marker in Chinese cabbage. (**a**) The candidate gene *Bra06H* was finally located in a region 15 Kb long between the A6S95 and A6S103 markers. (**b**) Association verification of MK in the F_2_ population constructed from a cross between G291 and ZHB. (**c**) Association verification of MK in commercial varieties of Chinese cabbage. Lines 1–15 are commercial varieties without trichomes, including Am4-1, Am3-1, Am5-2, Am9-1, Am94, AM30, Am54, Am72-1, Am10-1, Am56-1, QingyanChunbai 2, DegaoBaihe, Xibai 45, Y2, and L41. Lines 16-30 are commercial varieties with trichomes, including Xibai 8, Zhonghua 1, Qingyan CR20, QingyanXiabai 2, Guangdongzao, L46, Zhonghua 1, Gaokang 2, and Xibai 5. (**d**) Sequence alignment of the 5′-flaking region (upstream) of *BrGL1* in ZHB and G291.

**Figure 3 ijms-23-12721-f003:**
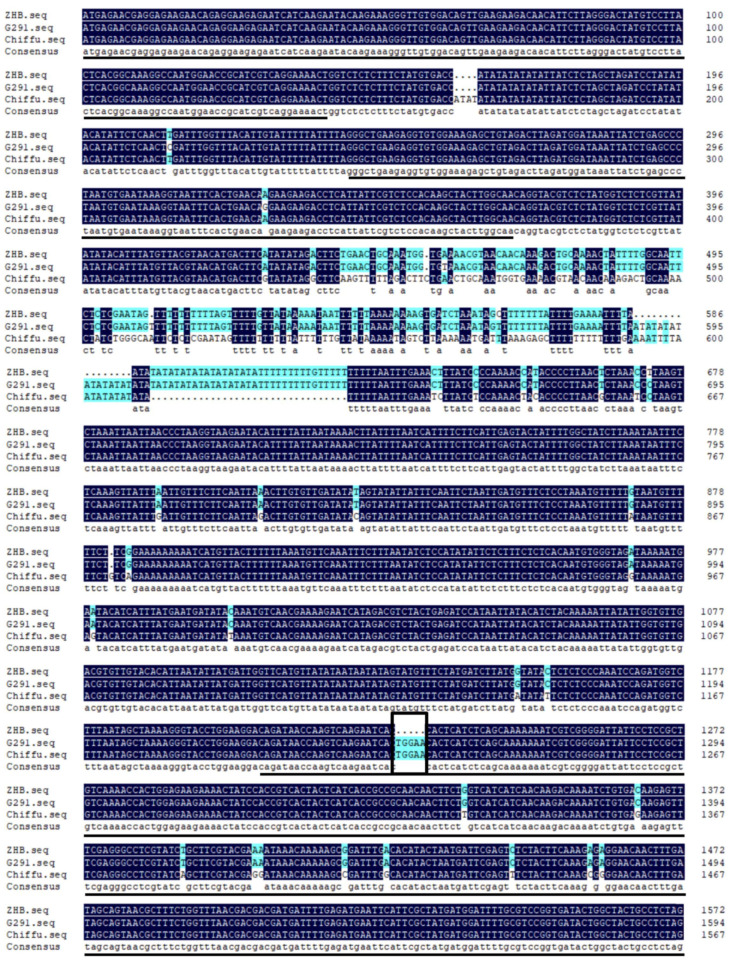
Sequence alignment analysis of the *BrGL1* gene between ZHB and G291. The three exons are underlined. The frame marks the 5-bp deletion site.

**Figure 4 ijms-23-12721-f004:**
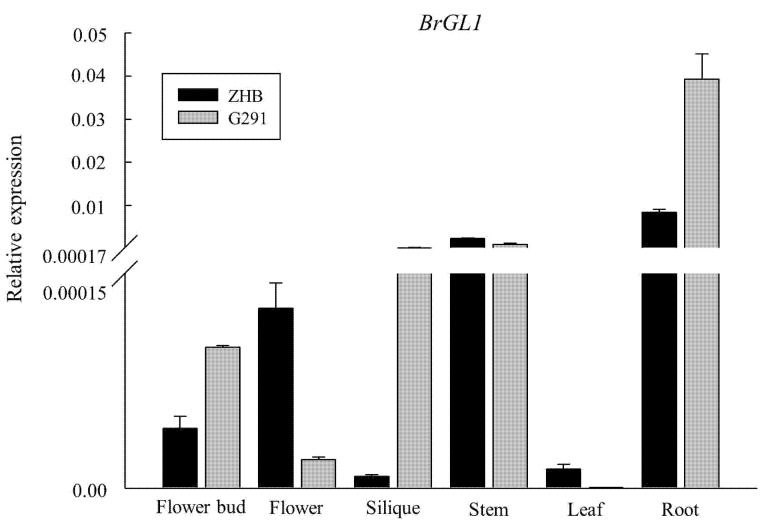
Relative expression of *BrGL1* in different tissues of the two parent lines of Chinese cabbage. The analysis was carried out by qRT-PCR. The expression levels of *BrGL1* were normalized to those of *BrACT1*, and the 2^−ΔΔCt^ method was used to calculate the expression levels of target genes in different tissues.

**Figure 5 ijms-23-12721-f005:**
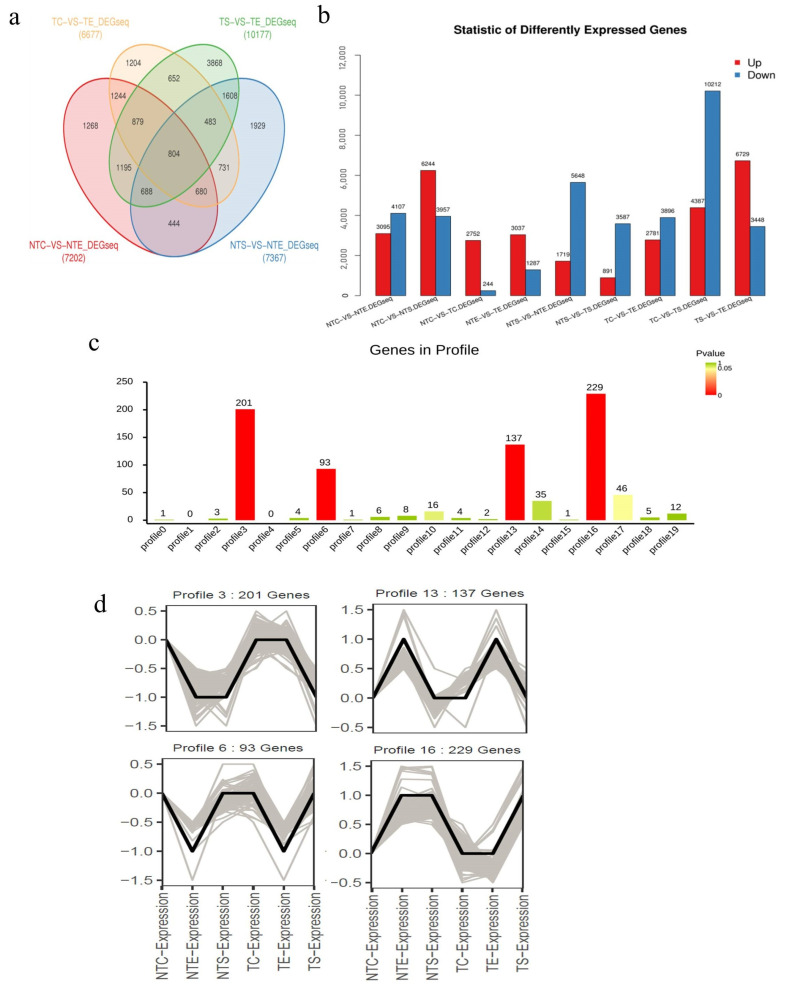
Analysis of differentially expressed genes (DEGs) identified by pairwise comparisons. (**a**) Venn diagram of the number of DEGs in the four paired comparisons. (**b**) The number of genes up- or down-regulated in the inoculated and non-inoculated samples with a Q-value < 0.001 and log_2_ fold change ≥ 1.0 or ≤ −1.0 are shown in the diagram. (**c**) Statistic of differentially expressed genes. The x-axis represents pairwise, and the y-axis represents the number of screened DEGs. The blue bar denotes the down-regulated genes, and the red bar denotes the up-regulated genes. (**d**) Cluster analysis of gene-expression patterns of DEGs. The 17,677 DEGs were grouped into 20 sub-clusters based on the common expression patterns by the OmicShare online tools (http://www.omicshare.com/tools (accessed on 21 October 2021)). Three sub-clusters reflected the significant expression trends of DEGs (*p*-value < 0.05). The gene number was marked on the top of each profile.

**Figure 6 ijms-23-12721-f006:**
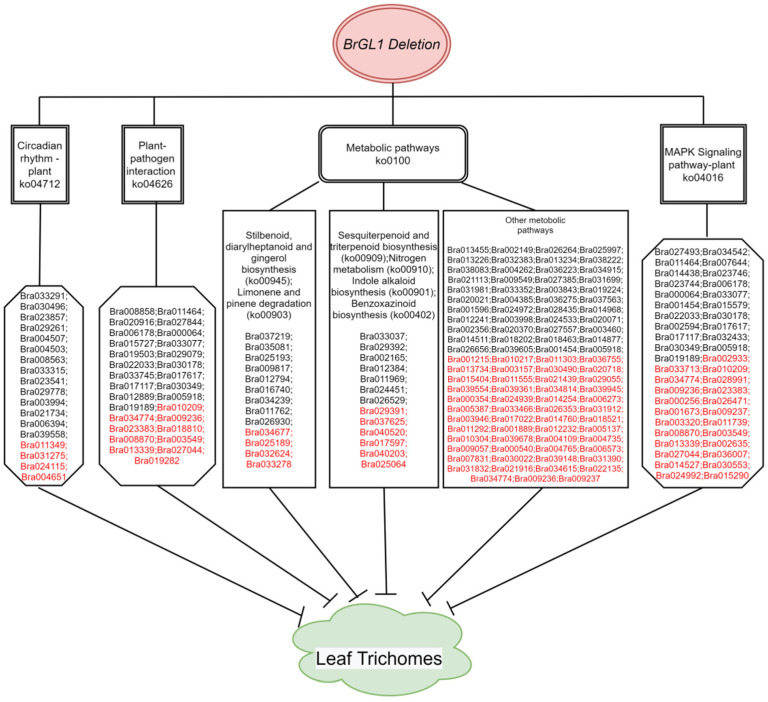
The hypothesis model of *BrGL1* regulating the development of leaf trichomes in Chinese cabbage. The candidate genes screened from transcriptome data are shown; the down-regulated genes are in black font, while the up-regulated genes are in red. Blocked segments represent suppression.

**Figure 7 ijms-23-12721-f007:**
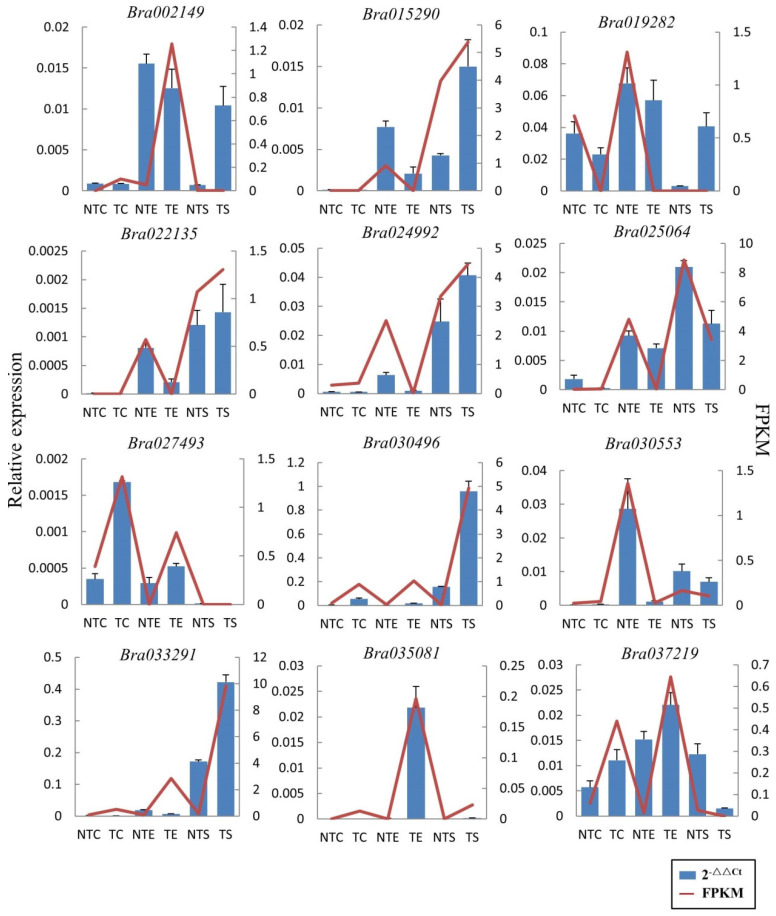
Validation of RNA-seq data by qRT-PCR. FPKM means reads per kb per million reads. The relative expression level was calculated by the comparative 2^−ΔΔCt^ method.

**Table 1 ijms-23-12721-t001:** χ^2^ test of leaf trichome traits in Chinese cabbage.

Item	Trichome	Glabrous	Total
Observed number	1075	381	1456
Theoretical number	1092	364	1456

χ^2^ = 0.2540 *p* < 0.05.

**Table 2 ijms-23-12721-t002:** Mapping of the candidate gene controlling trichomes in Chinese cabbage.

Chromosome	Peak Position (cM)	Left Marker	Right Marker	LOD Score	PVE (%)	Additive Effects	Dominant Effects
A06	318	A6S39	A6S38	31.2584	47.9685	−1.5546	−0.5968

**Table 3 ijms-23-12721-t003:** Statistics of RNA-seq data and map results.

Sample	Raw Reads (Mb)	Clean Reads (Mb)	Total Bases (Gb)	Map to Genes		Map to Genome	
				Total Mapping Ratio (%)	Uniquely Mapping Ratio (%)	Total Mapping Ratio (%)	Uniquely Mapping Ratio (%)
NTC_1	24.14	23.89	1.19	76.04	52.19	89.34	76.1
NTC_2	24.14	23.79	1.19	76.36	50.3	90.44	75.83
NTC_3	26.33	23.9	1.19	75.36	50.17	90.81	76.41
NTE_1	24.14	23.66	1.18	78.78	45.72	92.03	72.83
NTE_2	24.14	23.82	1.19	77.5	51.69	90.94	76.2
NTE_3	24.14	23.86	1.19	75.57	51.06	91.05	76.84
NTS_1	24.14	23.88	1.19	79.23	52.63	91.81	76.88
NTS_2	24.14	23.8	1.19	79.13	51.12	91.72	75.99
NTS_3	24.14	23.79	1.19	77.19	48.61	91.16	74.8
TC_1	24.14	24.03	1.2	77.02	53.27	90.9	77.6
TC_2	24.14	24.03	1.2	74.77	49.94	90.61	76.29
TC_3	24.14	24.05	1.2	77.68	53.54	91.09	77.44
TE_1	24.14	23.96	1.2	76.08	52.46	90.26	76.89
TE_2	24.14	24.01	1.2	76.64	52.66	90.62	77.31
TE_3	24.14	24.04	1.2	74.5	50.03	90.17	75.92
TS_1	24.14	24.05	1.2	80.17	51.24	92.17	75.95
TS_2	24.14	24.05	1.2	80.53	49.89	92.36	75.33
TS_3	24.14	23.98	1.2	80.75	50.41	92.29	75.57

**Table 4 ijms-23-12721-t004:** The most enriched KEGG pathways with significant differences at Q value ≤ 0.01 in the 9 comparison groups. The screened pathways are highlighted in red. The black dots indicate that the DEGs of each comparison group are enriched to the corresponding pathways displayed in the column.

KEGG Pathway (Q-Value < 0.01)	NTC vs. NTE	NTC vs. NTS	NTE vs. NTS	TC vs. TE	TC vs. TS	TE vs. TS	NTC vs. TC	NTE Vs. TE	NTS vs. TS
Biosynthesis of secondary metabolites	●	●		●	●	●		●	
Photosynthesis—antenna proteins	●	●	●		●	●		●	
Metabolic pathways	●	●				●		●	
Glutathione metabolism	●	●							
Pentose and glucuronate interconversions	●	●		●					●
Carbon metabolism	●	●			●				
Starch and sucrose metabolism	●	●	●			●			
Benzoxazinoid biosynthesis	●	●						●	
alpha-Linolenic acid metabolism	●	●			●	●			
Limonene and pinene degradation	●							●	
Plant hormone signal transduction	●	●		●		●		●	
Nitrogen metabolism	●								
Stilbenoid, diarylheptanoid and gingerol biosynthesis	●	●				●		●	
Cutin, suberine and wax biosynthesis	●	●							
Cysteine and methionine metabolism	●	●							
Alanine, aspartate and glutamate metabolism	●								
Phenylpropanoid biosynthesis	●	●							
Arginine and proline metabolism	●							●	
Fatty acid metabolism		●							
MAPK signaling pathway—plant		●		●		●		●	
Fatty acid biosynthesis		●							
Flavonoid biosynthesis		●				●			
Glycerolipid metabolism		●				●			
Carbon fixation in photosynthetic organisms		●							
Biotin metabolism		●							
Valine, leucine and isoleucine degradation		●							
Limonene and pinene degradation		●							
Biosynthesis of amino acids		●							
Indole alkaloid biosynthesis		●						●	
Citrate cycle (TCA cycle)		●							
Glyoxylate and dicarboxylate metabolism		●				●			
Phenylalanine metabolism		●							
Biosynthesis of unsaturated fatty acids		●							
Fatty acid degradation		●							
Amino sugar and nucleotide sugar metabolism		●							
Glycolysis / Gluconeogenesis		●							
Pyruvate metabolism		●							
Fructose and mannose metabolism		●							
Propanoate metabolism		●							
Peroxisome		●				●			
Sesquiterpenoid and triterpenoid biosynthesis		●		●					
Photosynthesis		●			●	●			
Steroid biosynthesis		●							
Circadian rhythm—plant				●				●	●
Plant-pathogen interaction				●			●	●	
Nitrogen metabolism				●				●	
Tryptophan metabolism						●			
Endocytosis							●		

## Data Availability

All sequence data were deposited in the NCBI Sequence Read Archive (SRA, http://www.ncbi.nlm.nih.gov/Traces/sra (accessed on 29 October 2021)) under accession numbers SAMN22568100, SAMN22568101, SAMN22568102, SAMN22568103, SAMN22568104, SAMN22568105 for NTC, NTE, NTS, TC, TE and TS, respectively.

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
