# Peer review of "Genetic and Transcriptome Analysis of Leaf Trichome Development in Chinese Cabbage (Brassica rapa L. subsp. pekinensis) and Molecular Marker Development"

_ijms, 2022, doi:10.3390/ijms232112721_

Round 1
Reviewer 1 Report
The present study was undertaken carefully and interesting results have been presented. But I have some concern as bellow.
1. As per the data generated and analysis done, the title is not representing the manuscript. The authors studied the genetics of trichome development and one marker found linked with it. They also RNA seq analysis for identification of candidates genes. The title may be revised.
2. In title, “spp.” may be write as “subsp.”
3. Line-12: “spp.” may be write as “subsp.”
4. Line-17: “Co-seperated” will be replaced with “Co-segregation”.
5. Line-28: Please arranged the keywords in alphabetical order.
6. The fine mapping may be deleted from the keywords.
7. Line-34, Line-50: Please expand the short forms as where comes first.
8. Line-60: The name of gene will be in italic.
9. :Line-65: The botanical name should be written in italic.
10. Line-68: The sentence will be written in past.
11. Line-70: correct the spp.
12. Line-76: Please make space between two words.
13. Line-94: The short form of QTL will be ok.
14. Line-99: Authors representing the QTL analysis. But it is just a co-segregation analysis.
15. Line-108: “The” will be replaced with “these”.
16. Line-113: “Construction” will be replaced with “development”.
17. Line-114: “An will be replaced with “A”.
18. Line 114-125: This section is not required as authors did not performed the QTL mapping for these agronomic traits.
19. Line-126: “Co-seperated” will be replaced with “Co-segregation”.
20. Line-134: 10ng/ul DNA is not enough for amplification. Please check it.
21. Line-130: The The PCR conditioning will also be added in the heading.
22. Line-139: The heading not representing the text.
23. LIne-147-155: Please correct as mentioned in the text.
24. Line-193: The heading nee to be changed.
25. .Line-194-202: This study is not performed here and data presented from the previous study. It may be deleted.
26. Line 214-215: The authors stated that the markers were developed for fine mapping. But F2 population is not suitable for fine mapping. Please correct it.
27. Line-220: “Co-seperated” will be replaced with “Co-segregation”.
28. Line-249: the name of gene should be in italic.
29. Line-268: The will be replaced with “these”.
30. Line-280: The “mixed” will be replaced with “pooled” .
31. The more information about importance of leaf trichome may be added in introduction.
32. Please also check the language. At some sections, too long sentences are written. Please see.
Please also see the attached file here for revision.

Reviewer 2 Report
The manuscript "Transcriptome Analysis of Leaf Trichome development in Chi- nese Cabbage (Brassica rapa L. ssp. pekinensis) and Molecular Marker Development" by Zheng et al. shows an interesting molecular analysis of the trichome presence in cabbage. They identified an indel that allowed to map of the putative gene responsible for the presence or absence of trichomes. Furthermore, they performed an RNAseq analysis to determine the genes involved in regulating the process.
The research is very well done. The only thing I did not find in the MS was the justification for seeing the QTL and the putative gene when it had been previously done by other authors cited in this research. The authors should clarify why they pursued the search for a QTL and a putative gene previously identified and characterized. Why not look directly for the gene before going for the long route? What was the advantage of following this long path? How do the results here compare to the previously published gene? It is the same, so?
The RNAseq analyses are placed as the backbone of the paper (see title); I think that drawing a model of the interactions could make the understanding of the findings easier.

Round 2
Reviewer 1 Report
Thanks to the authors for incorporation of comments for improvement of the manuscript. But still some minor corrections are there, please try to incorporate.
1. Caption of fig-2: The “fine mapping” will be replaced with “genetic mapping”.
2. Line-70, 74: The Arabidopsis will write in italics.
3. Please see the formatting of the text. There are some lines with gap. Please correct.
4. See the reference. Still it have duplicate numbering. Please correct it.
5. Line 508: The journal name will be in italic.
6. Line 530, 535, 545, 549: Arabidopsis will be in italic.
7. Reference no 24. Please correct the format.
8. Line 589, 593, 595, 598, 601, 603, 605, 608, 611, 614: The journal name will be in italic.
9. Line 585, 587, 610: The scientific name will be in italic.
10. Line 630: Correct the journal name.
